# Antifungal Activity of Cedrol from *Cunninghamia lanceolate* var. *konishii* against *Phellinus noxius* and Its Mechanism

**DOI:** 10.3390/plants13020321

**Published:** 2024-01-21

**Authors:** Wen-Wei Hsiao, Ka-Man Lau, Shih-Chang Chien, Fang-Hua Chu, Wen-Hsin Chung, Sheng-Yang Wang

**Affiliations:** 1Experimental Forest, College of Bio-Resources and Agriculture, National Taiwan University, Taipei 10617, Taiwan; hsiaoww@gmail.com; 2Department of Forestry, National Chung Hsing University, Taichung 40202, Taiwan; kakablauu@gmail.com; 3Experimental Forest Management Office, National Chung Hsing University, Taichung 40202, Taiwan; scchien@dragon.nchu.edu.tw; 4School of Forestry and Resource Conservation, National Taiwan University, Taipei 106217, Taiwan; fhchu@ntu.edu.tw; 5Department of Plant Pathology, National Chung Hsing University, Taichung 40202, Taiwan; wenchung@nchu.edu.tw; 6Special Crop and Metabolome Discipline Cluster, Academy Circle Economy, National Chung Hsing University, Taichung 40202, Taiwan; 7Agricultural Biotechnology Research Center, Academia Sinica, Taipei 11529, Taiwan

**Keywords:** *Cunninghamia lanceolata*, essential oil, brown root rot fungus, cedrol, antifungal activity, oxidative stress, apoptosis

## Abstract

*Phellinus noxius* is a highly destructive fungus that causes brown root disease in trees, leading to decay and death. In Taiwan, five prized woods—*Taiwania cryptomerioides*, *Calocedrus macrolepis* var. *formosana*, *Cunninghamia lanceolata* var. *konishii*, *Chamaecyparis formosensis*, and *Chamaecyparis obtusa* var. *formosana*—are known for their fragrance and durability. This study aims to explore the anti-brown-root-rot-fungus activity of *Cunninghamia lanceolata* var. *konishii* (CL) essential oil (CLOL) and its primary components, while also delving into their mechanisms of action and inhibition pathways. The essential oil (CLOL) from CL wood demonstrated significant efficacy against *P. noxius*, with an inhibitory concentration (IC_50_) of 37.5 µg/mL. Cedrol, the major component (78.48%) in CLOL, emerged as a potent antifungal agent, surpassing the reference drug triflumizole. Further assays with cedrol revealed a stronger anti-brown-root-disease activity (IC_50_ = 15.7 µg/mL) than triflumizole (IC_50_ = 32.1 µg/mL). Scanning electron microscopy showed deformation and rupture of fungal hyphae treated with CLOL and cedrol, indicating damage to the fungal cell membrane. Cedrol-induced oxidative stress in P. noxius was evidenced by increased reactive oxygen species (ROS) levels, leading to DNA fragmentation, mitochondrial membrane potential reduction, and fungal apoptosis through the mitochondrial pathway. Gel electrophoresis confirmed cedrol-induced DNA fragmentation, whereas TUNEL staining demonstrated increased apoptosis with rising cedrol concentrations. Moreover, protein expression analysis revealed cedrol-triggered release of cytochrome c, activation of caspase-9, and subsequent caspase-3 activation, initiating a caspase cascade reaction. This groundbreaking study establishes cedrol as the first compound to induce apoptosis in *P. noxius* while inhibiting its growth through oxidative stress, an increase in mitochondrial membrane permeability, and activation of the mitochondrial pathway. The findings offer compelling evidence for cedrol’s potential as an effective antifungal agent against the destructive brown root disease caused by *P. noxius*.

## 1. Introduction

The brown root rot fungus (*Phellinus noxius* (Corner) G. Cunn.) grows in tropical and subtropical regions [1,2,3]. It has a wide range of growth temperatures and can parasitize various economically important crops and landscape trees. The brown root rot fungus is an infective white-rot fungus capable of invading the bark and woody parts of trees, causing wood decay [4]. This damages the water transport and nutrient supply functions of the trees. Therefore, infected trees exhibit symptoms such as wilting leaves, the discoloration of leaves, the growth of brown fungal mycelium at the base of roots and stems [4], root tissue decay, and disintegration, ultimately leading to death. This loss of structural support can result in tree toppling, causing economic losses. Currently, the primary method for treating brown root rot involves removing infected trees and then using chemical agents for soil fumigation, a cumbersome and expensive process. To reduce the use of chemical agents, the development of natural remedies and confirmation of their efficacy is a future trend. The screening of new effective antifungal drugs can based on either (1) compounds with specific antifungal activity or (2) molecules aimed at enhancing the activity of existing antifungal drugs [5].

*Cunninghamia lanceolata* var. *konishii* (Taxodiaceae; CL), also known as Luan Da Shan or Taiwan cedar, is a sizable coniferous tree. Indigenous to the elevated forests of Taiwan, it thrives at altitudes ranging from 1300 to 2800 m. Alongside *Taiwania cryptomerioides* (TC), *Calocedrus macrolepis* var. *formosana* (CM), *Chamaecyparis formosensis* (CF), and *Chamaecyparis obtusa* var. *formosana* (CO), CL is collectively recognized as one of the “five precious woods of Taiwan”. The wood of CL is renowned for its straight and flawless grain, distinctive heartwood, and a pleasantly fragrant aroma. It finds applications in construction, furniture crafting, and shipbuilding. Mushrooms cultivated in the vicinity of CL have demonstrated antioxidant, anti-inflammatory, blood-sugar-lowering, and anti-aging properties. Furthermore, CL wood is a source of essential oils, typically extracted through steam distillation and water distillation. Essential oils, widely used in perfumes, cosmetics, and medical research, showcase anti-inflammatory, antioxidant, antibacterial, and antifungal attributes. Notably, the ethanol extract from CL wood exhibits resistance against termites [6] and inhibits wood-decaying fungi, including *Lenzites betulina*, *Trametes versicolor*, *Laetiporus sulphureus*, and *Gloeophyllum trabeum* [7]. Despite these findings, there is a notable gap in research regarding the impact of cedarwood essential oil on brown root rot fungus. Consequently, this study aims to explore the anti-brown-root-rot-fungus activity of CL wood essential oil and its primary components, while also delving into their mechanisms of action and inhibition pathways.

## 2. Results and Discussion

### 2.1. Comparing the Antifungal Activity of Taiwan’s Five Precious Woods

This study analyzes the activity of Taiwan’s five precious woods’ essential oils against *P. noxius*. The results, as shown in Table 1, indicate that the IC_50_ values for the antifungal activity of CL, TC, CM, CF, and CO are <50, <50, 72.3, 53.2, and 335.4 µg/mL, respectively. Among them, CL and TC exhibit better antifungal activity. Since TC has undergone various bioactivity tests conducted by many scholars [8,9,10,11] (the effective antifungal components of TC are ferruginol, T-cadinol, α-cadinol, and T-muurolol [12]) and there is only one study on the use of CL in antifungal research, the authors of this study chose CL wood essential oil for the study of antifungal activity.

Further antifungal activity tests were conducted using CL essential oil at concentrations of 10, 20, and 40 µg/mL. The IC_50_ of cedarwood essential oil against *P. noxius* was found to be 37.5 µg/mL. In comparison, a previous study on the main component of cinnamon leaf essential oil, *trans*-cinnamaldehyde, showed an IC_50_ of 116.0 µg/mL against *P. noxius* [13]. Therefore, the CL wood essential oil extracted in this study exhibits stronger antibacterial activity.

### 2.2. Composition Analysis of CLOL and Evaluation of Its Potent Antifungal Activity Compounds

In order to understand the active ingredients in cedarwood essential oil, gas chromatography–mass spectrometry (GC-MS) was used to identify the oil components. The results, as shown in Table 2, revealed that CL wood essential oil is primarily composed of cedrol, with a high content of 78.48%. Other components include α-cedrene, α-cadinol, β-cedrene, γ-eudesmol, globulol, and epicedrol, which are mainly terpenoid compounds. Therefore, it is speculated that cedrol may be the effective antibacterial component of CL wood essential oil. Subsequently, antifungal tests were conducted on the activity of cedrol against the root rot fungus.

In the aforementioned experiments, the IC_50_ of CL wood essential oil was determined to be 37.5 µg/mL (Table 3). Subsequently, antibacterial tests were conducted using cedrol at concentrations of 10, 20, and 40 µg/mL against the *P. noxius*, resulting in antibacterial indices of 45.4%, 50.3%, and 90.2%, respectively, with an IC_50_ of 15.7 µg/mL. Thiabendazole, a known inhibitor of phytophthora growth used as a fungicide, was also tested for its antibacterial activity, yielding an IC_50_ of 32.1 µg/mL. The test results indicate that cedrol is more effective at inhibiting the growth of phytophthora than thiabendazole, demonstrating the excellent antifungal activity of cedrol.

### 2.3. The Impact of Cedrol on the Morphology of P. noxius Hyphae

We analyzed the impacts of CL wood essential oil and cedrol on the morphology of *P. noxius*. As depicted in Figure 1 and Figure 2, with increases in the dosages of COCL and cedrol, the growth of fungal colonies decreased. Ethanol, used as a solvent for dissolving cedrol, does not reduce the growth of fungal colonies compared with the control group; in fact, it may even promote strain growth. This indicates that ethanol does not affect hyphal growth, which is consistent with the antibacterial index results. To observe changes in hyphal morphology, this study employed scanning electron microscopy (SEM) at a 2000× magnification for the essential oil and cedrol groups. As shown in Figure 3, with an increase in the dosage, there is a noticeable deformation of hyphae, and as the dosage increases, the deformation becomes more severe.

Further scrutiny at a 4500× magnification (Figure 4) revealed that the hyphae in the control group manifest normal growth (smooth surfaces and intact structures) and are in a growth phase. The ethanol group also presents a smooth appearance, signifying that ethanol does not impact the outward appearance of the fungal strain. Nevertheless, both the cedarwood essential oil and cedrol groups exhibit deformed hyphae. In the groups treated with 10 and 20 µg/mL of CL wood essential oil and cedrol, hyphae display wrinkling, which intensifies with increasing dosage, accompanied by curling. At 40 µg/mL, the hyphae in both the cedarwood essential oil and cedrol groups rupture, indicating damage to the fungal cell membrane and the release of cellular contents. The morphological alterations in fungi are associated with wood extracts, and these secondary metabolites act as antifungal substances, constraining fungal growth. Secondary metabolites impede fungal growth through various mechanisms, including disrupting cell membranes, suppressing the synthesis of fungal cell walls, inhibiting cell division, suppressing protein synthesis, and impairing mitochondrial function.

### 2.4. Cedrol Induces Oxidative Stress on P. noxius

Cedrol, classified as a sesquiterpene compound, exhibits antioxidant or pro-oxidant effects dependent on its structure, concentration, and the specific cell type involved [14] Reactive oxygen species (ROS) play a pivotal role in cellular processes such as signaling, metabolism, cell proliferation, differentiation, and aging [15]. Based on the SEM experimental findings discussed earlier, cedrol impedes the growth of *P. noxius* and induces membrane damage. Earlier investigations have demonstrated that in Candida albicans, disruption of the cell wall and accumulation of ROS result in cell death [16]. Therefore, this study seeks to examine whether cedrol treatment elevates the intracellular ROS content of *P. noxius* and consequently influences the strain’s growth.

The IC_50_ of cedrol against *P. noxius* is determined to be 15.7 µg/mL, which is equivalent to approximately 70.6 µM. Initially, co-treatment of *P. noxius* with 70 µM cedrol was attempted in this study. However, due to an insufficient quantity of mycelia obtained for the experiment, the concentration was subsequently reduced to 30 µM. As depicted in Figure 5, following treatment with 30 µM cedrol, a significant increase in intracellular ROS content in *P. noxius* cells was observed, indicating that cedrol induces oxidative stress in the fungus. The accumulation of a substantial amount of ROS can trigger programmed cell death. Consequently, further experiments will be conducted to elucidate the mechanism through which cedrol inhibits the growth of *P. noxius*.

### 2.5. Cedrol induces damage to P. noxius Genomic DNA (gDNA)

Cedrol has been demonstrated to induce damage to the genomic DNA (gDNA) of *P. noxius,* and *Candida albicans* triggers cell apoptosis through the accumulation of intracellular reactive oxygen species (ROS) [17,18]. DNA fragmentation, a biochemical manifestation of apoptosis, is assessable through gel electrophoresis to evaluate DNA degradation. As depicted in Figure 6, the gDNA of the EtOH (control) group exhibits significant brightness. In contrast, the brightness of the gDNA from *P. noxius* treated with 3 mM H_2_O_2_ shows a notable decrease, accompanied by a comet tail, indicating the destruction of double-stranded DNA. Moreover, treatment with DNase, which cleaves phosphodiester bonds on the DNA backbone through hydrolysis, in the EtOH group results in reduced overall DNA brightness and the appearance of a comet tail, signifying DNA fragmentation. The groups treated with 30, 50, and 70 µM cedrol all display comet tails, with the 50 µM cedrol group showing the most pronounced DNA fragmentation, indicating damage to the brown root rot fungus DNA. The comet tail in the 70 µM cedrol group is less pronounced than in the 50 µM cedrol group, with lower overall DNA brightness. This may be attributed to the high concentration of 70 µM cedrol inducing cell necrosis and resulting in relatively less intact DNA, which is consistent with the ROS results.

The results from DNA fragmentation experiments in *Bacillus subtilis* demonstrate a similar trend, with increasing damage levels and the occurrence of comet tails [19], aligning with the findings of this study. Generally, DNA fragmentation, when viewed using gel electrophoresis, results in a ladder-like pattern indicating DNA damage. Previous studies suggest that yeast ribosomes may lack or have fewer DNA linkages [20], resulting in a lack of classical DNA ladder patterns in electrophoresis. However, the absence of classic DNA ladder patterns does not impede DNA fragmentation from serving as a marker for fungal apoptosis [21].

### 2.6. Cedrol Induces Apoptosis in P. noxius

DNA fragmentation was evaluated through the TUNEL assay, wherein terminal deoxynucleotidyl transferase (TdT) binds dUTP to the 3′-OH positions at the ends of DNA breaks. Cellular apoptosis can be visualized by observing the fluorescence labeling of dUTP. Previous studies have demonstrated that subjecting yeast cells to 3 mM H_2_O_2_ for 200 min induces apoptosis, resulting in a 70% positive TUNEL phenotype [22]. In line with this approach, our study treated *P. noxius* with ethanol, 3 mM H_2_O_2_, and 30, 50, and 70 µM cedrol for 200 min. Figure 7 depicts the results, revealing that the ethanol control group exhibited some TUNEL fluorescence reaction. This reaction was attributed to ethanol inducing an increase in reactive oxygen species (ROS) in yeast cells, leading to oxidative stress [23,24]. Additionally, ethanol inhibits the biofilm formation of Candida albicans and Staphylococcus aureus, and with increasing ethanol concentration, propidium iodide (PI) staining intensity also increases, reflecting ethanol-induced damage to the cell membrane. However, the group treated with 3 mM H_2_O_2_ showed a significantly higher TUNEL fluorescence reaction compared with the ethanol group.

### 2.7. Cedrol Reduces the Mitochondrial Membrane Potential of P. noxius

Fungal cell apoptosis shares similarities with mammalian cell apoptosis; for example, both processes are regulated by mitochondria. The apoptotic process leads to heightened oxygen consumption and a decline in the mitochondrial membrane potential [25]. Investigating the impact of cedrol treatment on the mitochondrial membrane potential of *P. noxius* provides insights into whether cedrol triggers mitochondrial dysfunction, ultimately inducing apoptosis in the pathogen. JC-1, which accumulates on the negatively charged mitochondrial membrane, forms aggregates and exhibits red fluorescence, indicating a normal mitochondrial membrane potential and healthy cells. However, when the membrane potential decreases, JC-1 fails to bind to the negatively charged region on the mitochondrial membrane, dispersing within the cell and emitting green fluorescence.

As illustrated in Figure 8, although the mitochondrial membrane potential decreases in both the EtOH group and the 30 µM cinnamyl alcohol group, there are more healthy cells compared to the H_2_O_2_ group, which serves as a positive control for apoptosis. These groups exhibit similar changes in mitochondrial membrane potential as the 50 µM cinnamyl alcohol group. With increasing concentrations of cinnamyl alcohol, the intensity of green fluorescence emitted by JC-1 also rises, indicating that cedrol reduces the mitochondrial membrane potential of *P. noxius,* leading to mitochondrial dysfunction.

### 2.8. Pathway of Cedrol-Induced Apoptosis in P. noxius

Our findings demonstrate that cedrol treatment induces oxidative stress in *P. noxius*, resulting in DNA damage, increased mitochondrial membrane permeability, and a reduction in mitochondrial membrane potential, ultimately triggering apoptosis. Cell apoptosis can occur through intrinsic and extrinsic pathways. Cedrol stimulates a series of reactions in the cells of *P. noxius*, leading to apoptosis. Subsequent investigations aimed at understanding the apoptotic mechanism induced by cedrol treatment in *P. noxius* utilized the Western blot method to examine protein expression along the intrinsic pathway of apoptosis. The apoptosis network comprises proteins regulated by intricate cascade signals; studying the signaling pathways and mechanisms of action of apoptosis-related proteins serves as a method for controlling fungal infections.

Due to the increased mitochondrial permeability, AIF (Apoptosis-Inducing Factor) and cytochrome c are released from the mitochondria. Although yeast AIF shares 22% identity and 44% similarity with human AIF [26], this study did not detect the expression of the AIF protein. Comparing the cytochrome c sequences and activities of humans, horses, and *Rhizopus arrhizus*, similar activities were observed for their proteins [27]. As depicted in Figure 9 and Figure 10A, ethanol treatment resulted in a slight release of cytochrome c, whereas there was a significant increase in cytochrome c release after 24 h of treatment with 3 mM H_2_O_2_. Treatment with different concentrations of cedrol for 24 h also led to a significant increase in the protein expression of cytochrome c. Furthermore, compared with the positive control H_2_O_2_, treatment with 50 and 70 µM cedrol induced a more effective release of cytochrome c (*p* < 0.05), and treatment with 60 µM cedrol also significantly increased cytochrome c, caspase-9, and caspase-3 expressions in *P. noxius*. The apoptosis of *P. noxius* was induced by 3 mM H_2_O_2_. Treatment with different concentrations (30, 40, 50, 60, and 70 µM) of cedrol for 24 h. increased the release of cytochrome c (*p* < 0.01). Treatment with 60 µM cedrol resulted in the maximum beneficial release of cytochrome c, with results similar to those demonstrating a significant decrease in mitochondrial membrane potential with the 50 and 70 µM treatments.

After cytochrome c is released, it forms a complex with Apaf-1 and Procaspase-9 to create an apoptosome. However, this apoptotic process has not been confirmed in fungal cells. This study attempted to detect the protein expression of Apaf-1; however, it was not detected. Additionally, as illustrated in Figure 9 and Figure 10B, H_2_O_2_ significantly increased the expression of caspase-9. Treatment with different concentrations of cedrol induced the cleavage of Procaspase-9, activating it into caspase-9. Moreover, with increasing drug concentrations, there was a significant increase in caspase-9 expression.

Caspase-9, as an initiator of apoptosis, activates downstream proteins further. The subsequent detection of caspase-3 protein expression, as shown in Figure 9 and Figure 10C, revealed that H_2_O_2_ significantly increased caspase-3 expression. Treatment with 40, 50, and 60 µM cedrol also significantly increased caspase-3 expression. Caspase-3, an effector caspase in the apoptosis signaling pathway, when activated, initiates the cleavage of other death substrate proteins, inducing cell apoptosis and causing biochemical and morphological changes characteristic of apoptosis. In summary, the results of this study indicate that the cedrol treatment of *P. noxius* significantly increases the release of cytochrome c, activates caspase-9, and leads to the activation of caspase-3, triggering a cascade of caspases and initiating the apoptosis process.

## 3. Materials and Methods

### 3.1. Preparation Essential Oil from Cunninghamia lanceolate var. konishii Wood and Its Composition Analysis

The CL wood used in this study was provided by Dr. Min Jay Chung (Experimental Forest Management Office of National Taiwan University, Nantou County, Taiwan). CL wood was extracted using the steam distillation method and heating for 6 h. Taking advantage of the immiscibility of essential oil and water, the upper layer of essential oil was collected using a glass dropper and stored at 4 °C for subsequent composition analysis and antifungal activity analysis.

Gas chromatography–mass spectrometry (GC-MS, Trace GC Ultra coupling with ITQ 900 mass system, Thermo Co., Saint Louis, MO, USA) was employed for the analysis of the essential oil components. The essential oil was diluted to 0.1% with ethyl acetate, and 1 µL was injected into GC-MS for analysis. The injection port temperature was set at 240 °C. The initial temperature was 40 °C, increased at a rate of 3 °C/min to 180 °C, then ramped up to 280 °C at a rate of 20 °C/min, and finally held at 280 °C for 5 min. Helium was used as the carrier gas with a flow rate of 1.0 mL/min, and the column used was a DB-5MS (30 m × 0.25 mm i.d., 0.25 film, J & W Scientific, Folsom, CA, USA). The ionization voltage of the mass spectrometer was 70 eV, the ionization temperature was 250 °C, and the scanning range was set at 50–650 *m*/*z*. The obtained mass spectra fragments were compared with the Wiley/NBS and NIST (National Institute of Standards and Technology, Gaithersburg, MA, USA) databases. Further confirmation of the compounds was carried out using standard samples.

### 3.2. Antifungal Assay

The antifungal assay was performed in this study to evaluate the antifungal activity of essential oils and cedrol. The brown root rot fungal strain, *P. noxius,* was isolated from *Ficus macrocarpa* by Dr. Wen-Wei Hsiao. The strain (Exfo00145) was stored at the Plant Pathology and Microbiology Center, National Taiwan University Experimental Forest. The isolate originated from the Tropical Arboretum at the National Taiwan University Experimental Forest, Lutsao, Taiwan. Antifungal assays were performed three times and the data were averaged. Different concentrations of CLOL or cedrol were added to sterilized potato dextrose agar (PDA) to give 10, 20, and 40 µg/mL concentrations of essential oil and cedrol. Triflumizole (40 µg/mL) was used as the reference drug. The testing plates were incubated at 27 ± 2 °C. When the mycelium of the fungi reached the edge of the control plate the antifungal index was calculated as follows: antifungal index (%) = (1 − Da/Db) × 100, where Da: diameter of growth zone in the experimental dish (cm) and Db: diameter of growth zone in the control dish (cm).

### 3.3. Examining Fungal Hyphal Growth through Scanning Electron Microscopy (SEM)

In this experiment, the SEM was employed to observe the hyphal growth morphology. The solid dilution method was utilized, where PDA was sterilized and then cooled to 50 °C. Essential oil or cedar alcohol was dissolved in 99.5% ethanol and added to the culture medium. In the control group, alcohol with a total volume concentration of 1% was added. After the culture medium cooled and solidified, fungal plugs were inoculated and incubated at 25 ± 1 °C. When the fungal hyphae covered the culture dish in the control group, three 3 mm fungal plugs were taken from each group. These plugs were placed in 1.5 mL microcentrifuge tubes, and 1 mL of 2.5% glutaraldehyde was added as a fixative. The tubes were immersed in a dark environment at 4°C for 8–12 h. Sequential dehydration was performed using ethanol at different concentrations (30%, 40%, 50%, 70%, and 90%) for 5 min each, followed by three immersions in 95% and 99.5% ethanol for 5 min each. After dehydration, a critical point drying machine (Quorum E3100) was used to replace moisture with CO_2_ for drying. Once dried, the samples were taped onto a holder and coated with a gold film using a sputter coater (Quorum SC7620) at 15 mA for 100 s. The surface morphology of the fungal hyphae was then observed using an SEM (HITACHI S-3400N, Hitachi, Tokyo, Japan).

### 3.4. DCFH-DA Fluorescence Staining

Using a liquid cultivation method, after sterilizing and cooling MEB (Malt Extract Broth) to room temperature, each group has 150 mL of MEB poured in and 3 pieces of microbial blocks (3 mm) are inoculated. The treatment group is subjected to drug treatment with 30 µM cedar alcohol, while the control group is treated with 1.5 mL of alcohol. After incubating for 4 days at 25 ± 2 °C, the DCFH-DA light-avoidance reaction is initiated for 30 min. Filaments are collected and filtered, followed by rapid freezing and drying with liquid nitrogen, resulting in a powdered form. One gram of the sample is weighed and added to 1 mL of Tris lysis buffer. Centrifugation is performed at 9.0× *g* at 4 °C for 5 min and the supernatant is collected. The supernatant is filtered through a 0.45 µm filter, and 200 µL of the filtered supernatant is taken into a black 96-well plate. A multifunctional microplate analyzer is used to measure the fluorescence absorption of the supernatant (Ex/Em: 504/529).

### 3.5. DNA Electrophoresis

In this experiment, a liquid cultivation method was employed. After sterilizing and cooling MEB to room temperature, each group had 150 mL of MEB poured in and 3 pieces of microbial blocks (3 mm) were inoculated. The cultures were incubated at 25 ± 2 °C for 4 days. Subsequently, 150 µL of EtOH, 3 mM H_2_O_2_, and 30, 50, and 70 µM cedar alcohol were added separately. After 24 h of drug treatment, the mycelia were collected and filtered to remove the culture medium. Samples weighing 200 mg were taken and the total DNA of the brown root rot fungus was extracted and purified according to the instructions of the PrestoTM Mini Gdna Yeast Kit (Geneaid Biotech, Taipei City, Taiwan) and stored at −80 °C. DNA electrophoresis utilized agarose gel electrophoresis. Initially, the DNA concentration was measured using a spectrophotometer, and DNA was prepared at a concentration of 80 ng/µL. A 1% agarose gel was prepared using 1× TAE and SeaKem^®^ LE Agarose (Lonzc, Laussane, Switzerland). During gel preparation, Green DNA Dye (Infinigen, Salinas, Puerto Rico) was added for internal staining. Since the dye is a positively charged fluorescent dye, during electrophoresis the gel displayed uneven brightness due to the electric field relationship. However, this approach provided better resolution. After solidification of the agarose gel, it was placed in the electrophoresis tank, 1 × TAE was added as a buffer solution, and a voltage of 80 volts was applied. Smaller molecules move faster through the gel matrix pores, facilitating the separation of different lengths of DNA fragments. Finally, the DNA labeling amount was detected and analyzed using a ChemiDoc XRS+ cold light imaging system (Bio-Rad Laboratories, Inc., Hercules, CA, USA).

### 3.6. TUNEL Staining

The procedure involved first cultivating through a liquid cultivation method for 4 days. Subsequently, the application of drugs included a 200 min treatment with 3 mM H_2_O_2_ and 30, 50, and 70 µM cedar alcohol, with ethanol used as the control group. Fungal hyphae were extracted into 1.5 mL microcentrifuge tubes, the culture medium was removed, and 200 µL of 20% KOH was added for a 10 min treatment. Staining and fixation of the fungal hyphae were performed according to the Cell Meter™ TUNEL Apoptosis Assay Kit (AAT Bioquest, Pleasanton, CA, USA). Following this, the fungal hyphae were placed on glass slides, covered with a coverslip, and pressure was applied to ensure even dispersion of the hyphae, preventing a decrease in resolution due to excessive thickness. Finally, using a dual-scanning spectral laser confocal microscope (Olympus, Tokyo, Japan), the 3′-OH positions of DNA fragmentation were observed, as they displayed green fluorescence.

### 3.7. Mitochondrial Membrane Potential Analysis

The brown root rot fungus was cultured using a liquid cultivation method for 4 days. After 24 h of drug treatment, fungal hyphae were extracted into 1.5 mL microcentrifuge tubes, the culture medium was removed, and a 10 min pre-treatment with 200 µL of 20% KOH was applied. Subsequently, staining and fixation of the fungal hyphae were carried out following the instructions of the JC-1 dye (Abbkine, Atlanta, GA, USA). The fungal hyphae were then placed on glass slides and covered with a coverslip to evenly disperse the hyphae. Finally, fluorescence microscopy was employed to observe the cellular staining.

### 3.8. Protein Expression Analysis

In this experiment, a liquid cultivation method was employed with the concentration of the culture medium adjusted according to the MEB cultivation instructions. After sterilization and cooling to room temperature, 150 mL of MEB was poured into each sample and 3 pieces of microbial blocks (3 mm) were inoculated. The cultures were incubated at 25 ± 2 °C for 4 days and, subsequently, 150 µL each of EtOH, 3 mM H_2_O_2_, and 30–70 µM cedrol were added separately. After 24 h of drug treatment, fungal hyphae were collected by filtration, rapidly freeze-dried with liquid nitrogen, ground into a powder using a mortar in liquid nitrogen, and then 0.5 g was weighed. Next, 300 µL of T-PER™ Tissue Protein Extraction Reagent (Thermo, Waltham, MA, USA) was added. The samples were then subjected to 30 s of sonication using an ultrasonic cell disruptor, followed by centrifugation at 20× *g*, 4 °C for 5 min. The supernatant obtained represented the total protein and was stored at −80 °C for further analysis.

To ensure consistent protein concentrations across experimental groups, bovine serum albumin (BSA, Gibco, Grand Island, NY, USA) was used as a standard for protein concentration determination. BSA was prepared at concentrations of 0.1, 0.2, 0.3, 0.4, and 0.5 mg/mL. Sample proteins were diluted 20 times, with a volume of 50 µL, and tested in quadruplicate. Both the prepared BSA and sample proteins (10 µL each) were added to a 96-well plate, followed by the addition of 200 µL of Protein Assay Dye Reagent Concentrate (Bio-Rad Laboratories, Inc., Hercules, CA, USA) for 5 min for the light-avoidance reaction. Subsequently, ELISA was used to measure the absorbance values at a wavelength of 595 nm. The average absorbance values for different concentrations of BSA were used to calculate a linear regression equation, with an R^2^ value of at least 0.99. The average absorbance values of the sample proteins were then inserted into the regression equation and multiplied by the dilution factor to determine the protein concentration (µg/µL). Following this, the protein concentration for each group was adjusted to 60 µg/µL and 1/5th of the total volume of 5X Protein Sample Dye was added. The mixture was homogenized and reacted at 95 °C for 5 min to denature the proteins, and the denatured proteins were stored at −80 °C.

The protein gel electrophoresis analysis utilized sodium dodecyl sulfate-polyacrylamide gel electrophoresis (SDS-PAGE). When voltage is applied, negatively charged SDS wraps around proteins, causing them to move toward the positive pole. In the polyacrylamide structure, proteins separate based on their molecular weight, with larger proteins moving more slowly and smaller proteins moving faster. This experiment employed 10–15% SDS-PAGE gels for electrophoresis, and the gel composition is outlined in Appendix A. Electrophoresis was conducted with an upper gel voltage of 30–40 volts and a lower gel voltage of 80–100 volts. Due to the fragility of the gel material, SDS-PAGE-separated proteins were transferred onto polyvinylidene difluoride (PVDF) membranes. PVDF membranes offer good flexibility and toughness, facilitating handling and long-term storage. The protein transfer conditions from the SDS-PAGE gel to the PVDF membrane included a current of 300 mA, a transfer time of 60–120 min, and the experimental process was maintained in a cooled state.

Since PVDF membranes are porous and can adsorb proteins, non-target proteins or antibodies may bind to the membrane, leading to increased background or poor antibody specificity. Therefore, after the transfer, the PVDF membrane was blocked using a Western blot blocking buffer (Blockpro 1 Min Protein-Free Blocking Buffer, Youhe, Taiwan) at room temperature with gentle shaking for 1 min to prevent nonspecific binding. After blocking, the membrane was washed three times with TBS-T buffer for 5 min each time. Subsequently, the primary antibody was added and the membrane was shaken at 4 °C for 12–16 h. The primary antibodies used are listed in Appendix A. Following incubation, the membrane was washed three times with TBS-T buffer for 5 min each time. Then, the corresponding secondary antibody for the species of the primary antibody was added and the membrane was shaken at room temperature for 2 h. Afterward, the membrane was washed three times with TBS-T buffer for 5 min each time. Finally, chemiluminescent reagent (WesternBright™ ECL, Advansta Inc., San Jose, CA, USA) was applied to induce chemiluminescence in the target proteins. The protein expression levels were detected and analyzed using a ChemiDoc XRS+ cold light imaging system (Bio-Rad Laboratories, Inc., USA).

### 3.9. Statistical Analysis

The experimental results are expressed as mean ± standard deviation (Mean ± S.D.). All experiments were conducted in triplicate, and statistical analysis was performed using GraphPad Prism 8 software with one-way analysis of variance (One-Way ANOVA). Turkey’s post hoc test was applied to analyze all data, and *p* < 0.05 was considered indicative of significant differences in the data.

## 4. Conclusions

The study investigates the inhibitory effects of *C. lanceolata* essential oil on the growth of *P. noxius*, the causal agent of root rot disease. Through GC-MS analysis, the active compound in the essential oil was identified as cedrol, which constituted 78.48% of the oil. The IC_50_ of cedrol against the pathogen was determined to be 15.7 µg/mL, demonstrating stronger antifungal activity compared with the known Phytophthora inhibitor triflumizole. SEM observations revealed that both the essential oil and cedrol caused morphological deformations in the hyphae of the pathogen. With increasing concentrations of the treatment, hyphal rupture, cell membrane damage, and release of cellular contents were observed. As many cellular signaling pathways are associated with ROS, the study examined the ROS levels in *P. noxius*. Treatment with 30 µM cedrol increased intracellular ROS, leading to oxidative stress. Accumulation of oxidative stress can induce cell apoptosis, and gel electrophoresis of gDNA indicated damage induced by cedrol. The TUNEL assay further confirmed DNA fragmentation, with increased fluorescence intensity correlating with higher cedrol concentrations, reflecting the induction of apoptosis in *P. noxius* cells. To elucidate the mechanism of cedrol-induced apoptosis, the study assessed the mitochondrial membrane potential, revealing a decrease and increased permeability after 6 h of cedrol treatment. Protein expression analysis confirmed that the loss of mitochondrial membrane potential significantly triggered the release of cytochrome c, subsequently activating caspase-9, which further activated caspase-3, initiating the apoptotic pathway. In summary, cedrol, as a natural compound, holds potential for development as a natural antimicrobial agent. Its application in soil fumigation could reduce the reliance on chemical pesticides, contributing to sustainable development and a circular economy (Figure 11). This study provides the first evidence that *C. lanceolata* essential oil, particularly its main component cedrol, can combat *P. noxius*, offering insights into the apoptotic mechanisms involved in the pathogen.

## Figures and Tables

**Figure 1 plants-13-00321-f001:**
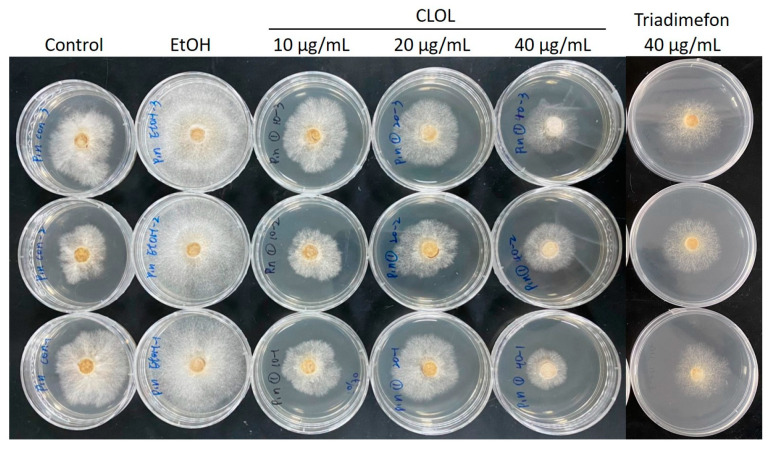
Inhibition effects of 10, 20, and 40 µg/mL essential oil of *Cunninghamia lanceolate* var. *konishii* against *P. noxius.* Control row initiates tributed experiments of control; EtOH row initiates tributed vehicle control (ethanol only); CLOL rows indicate different dosages of essential oils treatment; Tridimefon row indicates tributed positive control treatment.

**Figure 2 plants-13-00321-f002:**
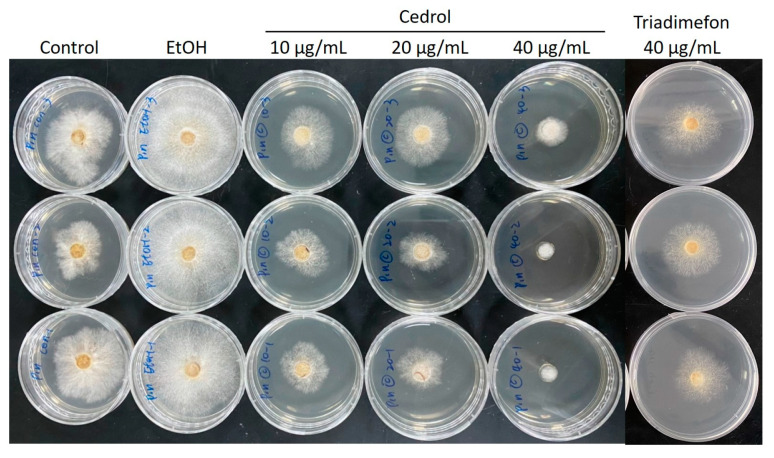
Inhibition effects of 10, 20, and 40 µg/mL cedrol against *P. noxius.* Control row initiates tributed experiments of control; EtOH row initiates tributed vehicle control (ethanol only); CLOL rows indicate different dosages ofcedrol treatment; Tridimefon row indicates tributed positive control treatment.

**Figure 3 plants-13-00321-f003:**
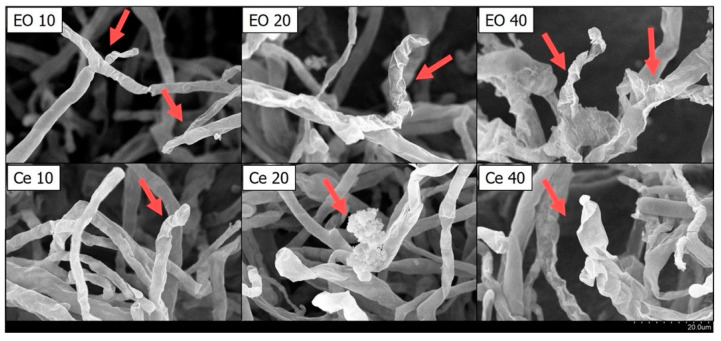
Effect of the essential oil of *Cunninghamia lanceolate* var. *konishii* and cedrol on *P. noxius* mycelium morphology viewed under SEM at 2000×. EO10: 10 µg/mL essential oil; EO20: 20 µg/mL essential oil; EO40: 40 µg/mL essential oil; Ce 10: 10 µg/mL cedrol; Ce 20: 20 µg/mL cedrol; Ce 40: 40 µg/mL cedrol. The red arrows indicates that the hyphae are damaged by essential oil or cedrol and become wrinkled.

**Figure 4 plants-13-00321-f004:**
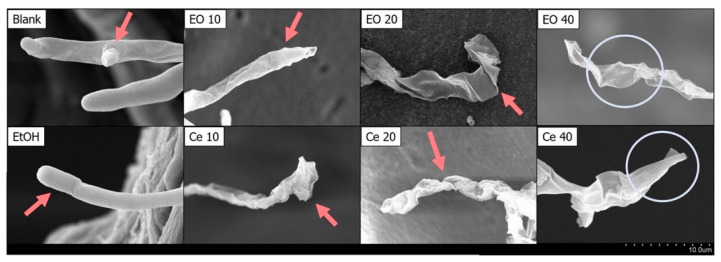
Effect of the essential oil of *Cunninghamia lanceolate* var. *konishii* and cedrol on *P. noxius* mycelium morphology viewed under SEM at 4500×. Blank: no treatment; EtOH: 99.5% ethanol; EO10: 10 µg/mL essential oil; EO20: 20 µg/mL essential oil; EO40: 40 µg/mL essential oil; Ce 10: 10 µg/mL cedrol; Ce 20: 20 µg/mL cedrol; Ce 40: 40 µg/mL cedrol. The red arrows indicates that the hyphae are damaged by essential oil or cedrol and become wrinkled.

**Figure 5 plants-13-00321-f005:**
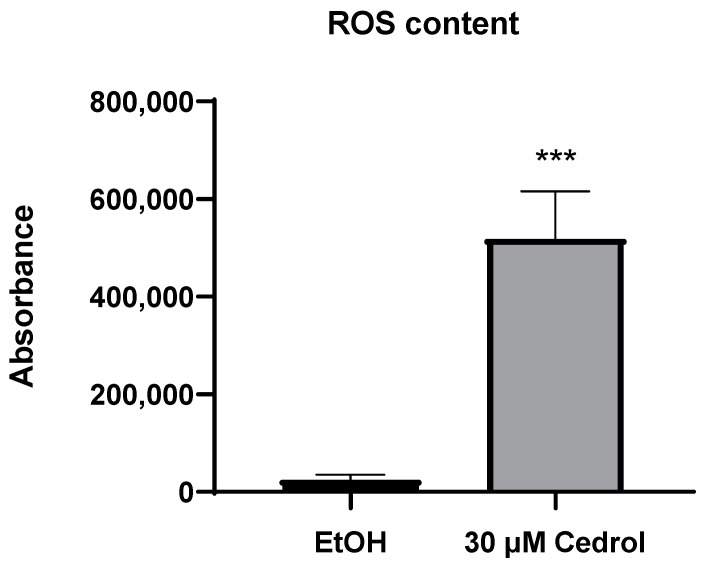
ROS content of 30 µM cedrol on *P. noxius.* The data are presented as mean ± S.D. (n = 3). Statistical analysis using Student’s *t*-test compared with control (EtOH). *** *p* < 0.001.

**Figure 6 plants-13-00321-f006:**
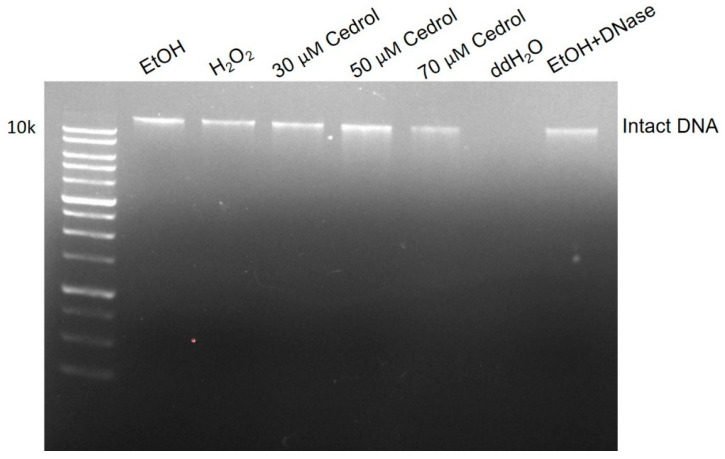
DNA fragmentation in *P. noxius* assessed via gel electrophoresis. Apoptosis in *P. noxius* was induced using 3 mM H_2_O_2_ or treatment with different concentrations (30, 50, and 70 µM) of cedrol for 24 h. Lane 1: Marker, Lane 2: EtOH, Lane 3: 3 mM H_2_O_2_, Lane 4: 30 µM cedrol, Lane 5: 50 µM cedrol, Lane 6: 70 µM cedrol, Lane 7: ddH_2_O, Lane 8: EtOH + DNase.

**Figure 7 plants-13-00321-f007:**
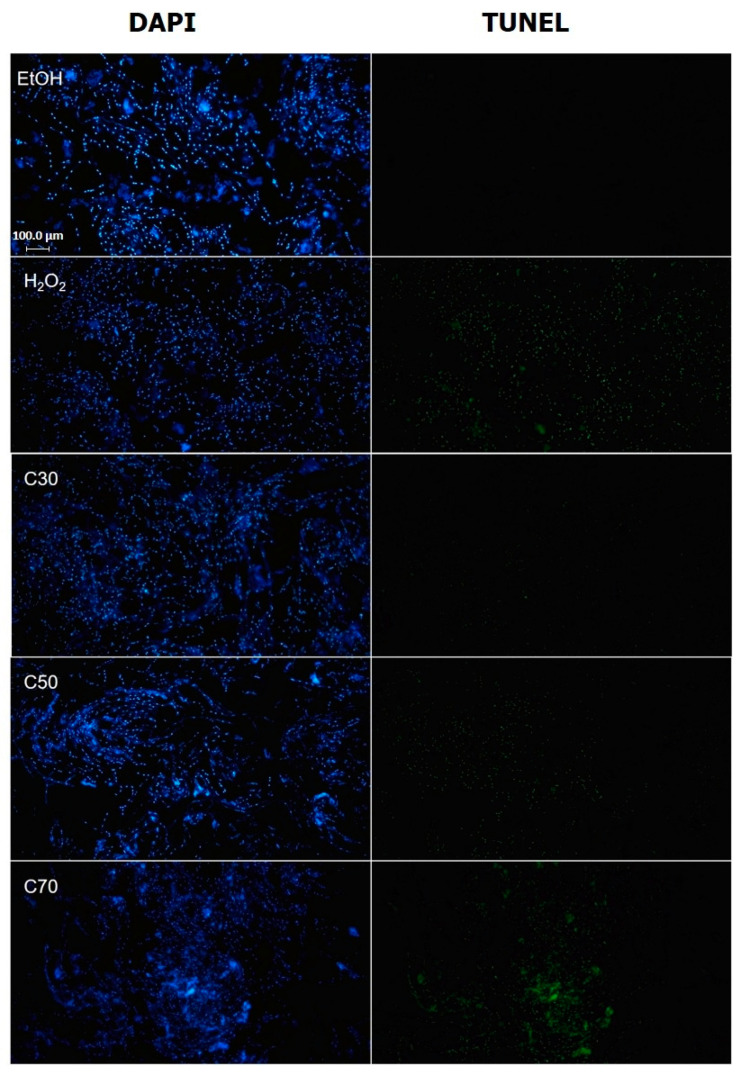
Representative images of the terminal deoxynucleotidyl transferase dUTP nick-end labeling (TUNEL) assay in *P. noxius* cultured with EtOH, H_2_O_2_, and cedrol. Terminal deoxynucleotidyl transferase (TdT) reacts with fluorescein-labeled dUTP to attach uridine to the 3′-hydroxyl (3′OH) terminus in DNA strand breaks. Apoptosis in *P. noxius* was induced via 3 mM H_2_O_2_ or treatment with different concentrations (30, 50, and 70 µM) of cedrol for 200 min.

**Figure 8 plants-13-00321-f008:**
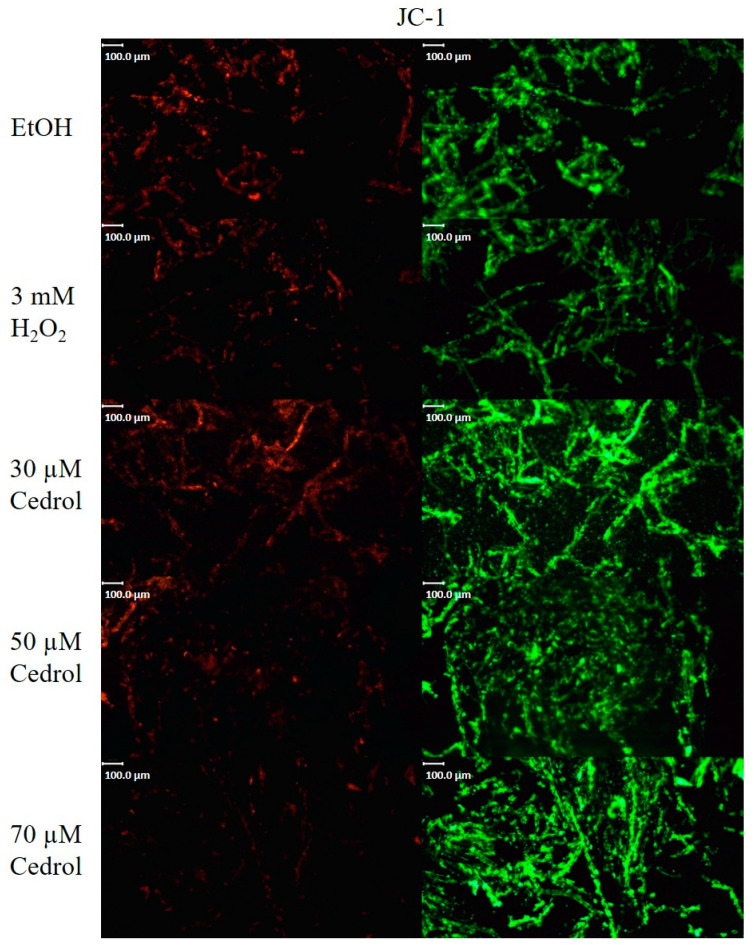
Mitochondria membrane potential in *P. noxius* with EtOH, H_2_O_2_ and cedrol. *P. noxius* stained with JC-1 and the green fluorescence following exposure to apoptosis. Apoptosis in *P. noxius* was induced by 3 mM H_2_O_2_, treatment with different concentration (30, 50, 70 µM cedrol) of cedrol for 6 h.

**Figure 9 plants-13-00321-f009:**
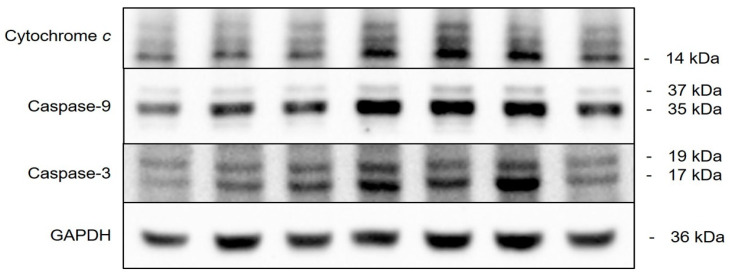
Cytochrome c, caspase-9, and caspase-3 expressions in *P. noxius* cultured with cedrol. Apoptosis in *P. noxius* was induced using 3 mM H_2_O_2_ or treatment with different concentrations (30, 40, 50, 60, and 70 µM) of cedrol for 24 h.

**Figure 10 plants-13-00321-f010:**
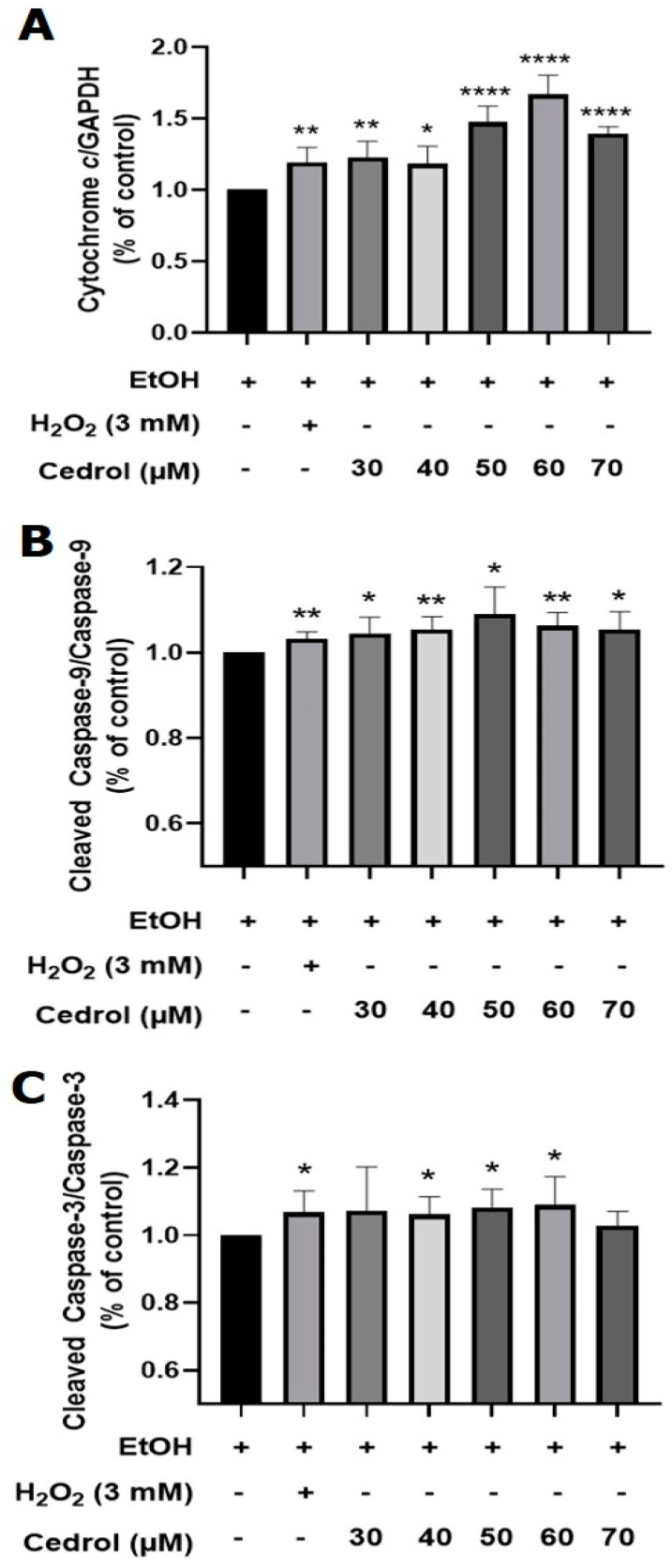
Cytochrome *c* (**A**), caspase-9 (**B**), and caspase-3 (**C**) expressions in *P. noxius* cultured with cedrol. Apoptosis in *P. noxius* was induced using 3 mM H_2_O_2_ or treatment with different concentrations (30, 40, 50, 60, and 70 µM) of cedrol for 24 h. The data are presented as mean ± S.D. (n ≥ 3). Statistical analysis using Student’s *t*-test compared with control (EtOH). * *p* < 0.05, ** *p* < 0.01, and **** *p* < 0.0001.

**Figure 11 plants-13-00321-f011:**
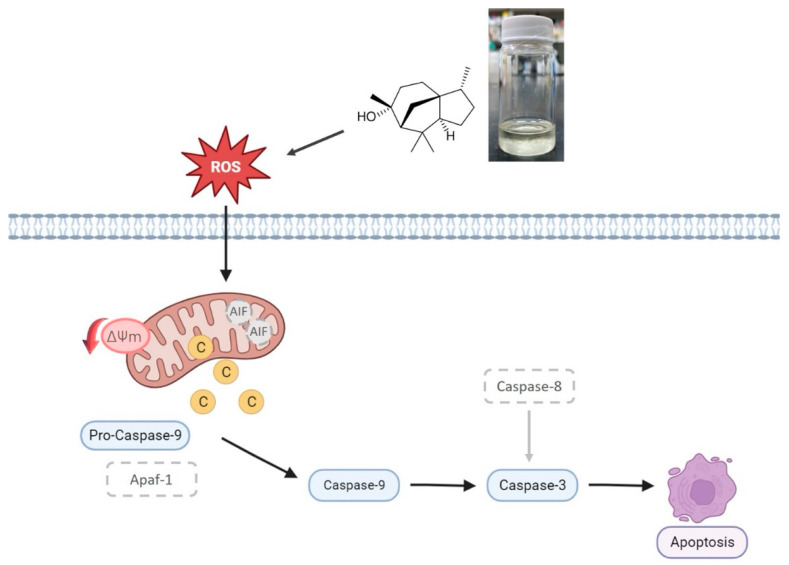
The apoptosis pathway in *P. noxius* was induced by cedrol.

**Table 1 plants-13-00321-t001:** Antifungal activity of Cunninghamia lanceolate var. konishii (CL), Taiwania cryptomerioides (TC), Calocedrus macrolepis var. formosana (CM), Chamaecyparis formosensis (CF), and Chamaecyparis obtusa var. formosana (CO) against *P. noxius*.

µg/mL	CL	TC	CM	CF	CO
50	57.7%	53.8%	35.8%	48.5%	15.4%
100	69.2%	65.4%	67.7%	72.3%	11.5%
200	76.9%	76.9%	100%	100%	26.9%
400	80.8%	80.8%	100%	100%	61.5%
IC_50_	<50	<50	72.3	53.2	335.4

**Table 2 plants-13-00321-t002:** GS-MS analysis of essential oil of *Cunninghamia lanceolate* var. *konishii*.

RT	KI	Concentration%	Constituent	Identification
21.16	1191	0.70	α-Terpineol	KI/MS/ST
31.12	1410	8.66	α-Cedrene	KI/MS/ST
31.44	1418	2.46	β-Cedrene	KI/MS/ST
31.91	1430	0.38	*cis*-Thujopsene	KI/MS
34.91	1501	0.36	Cuparene	KI/MS/ST
35.04	1504	0.43	α-Chamigrene	KI/MS/ST
38.39	1589	1.06	Globulol	KI/MS
38.97	1604	78.48	Cedrol	KI/MS/ST
39.40	1617	1.05	*epi*-Cedrol	KI/MS/ST
39.75	1627	1.59	γ-Eudesmol	KI/MS/ST
40.61	1651	2.87	α-Cadinol	KI/MS/ST
	Total	98.04		

RT: retention tims. KI: Kovats retention index on a DB-5MS column in reference to *n*-alkanes. ST: Authentic standard compounds. MS: NIST and Wiley libraries literature.

**Table 3 plants-13-00321-t003:** Antifungal activity of essential oil of *Cunninghamia lanceolate* var. *konishii* and its main compound cedrol against *P. noxius*.

Concentration (µg/mL)	CL Wood Essential Oils	Cedrol
10	39.1%	45.4%
20	34.0%	50.3%
40	52.6%	90.2%
IC_50_	37.5	15.7

## Data Availability

All data generated or analyzed during this study are included in this published article and Appendix A.

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
