# Peer review of "Antifungal Activity of Cedrol from *Cunninghamia lanceolate* var. *konishii* against *Phellinus noxius* and Its Mechanism"

_plants, 2024, doi:10.3390/plants13020321_

Round 1
Reviewer 1 Report
Comments and Suggestions for Authors
A review of the manuscript entitled "Antifungal activity of cedrol from Cunninghamia lanceolate var. konishii against Phellinus noxius and its mechanism" by Hsiao et al. I found the manuscript interesting and valuable. The research was done correctly. The introduction is very brief. Only 7 items of literature are cited! Many valuable papers showing the effectiveness of different essential oils to control fungi on different plants can be reviewed. Items 8-11 are not quoted in the text. The discussion of the results is also very sparse. It only covers 16 items of literature. I believe that the scope of the study was broad and any result of the work needs to be discussed with the available literature. In the chapter Material and methods, no literature item was cited in the description of the methodology.
I recommend that the paper be accepted for publication with Major revision.
Specific comments:
1. The abstract should include brief information about the scope of the research work and the research methods used.
2. Keywords: repeat the information contained in the title of the paper. This should be amended.
3. L84. The title of section 2.1. reads like a summary. The title should be changed e.g. Antifungal activity of selected essential oils. The material and methods section should describe the experiment!
4. L106. Again the title of section 2.2. is a conclusion?
5. L116. GS-MS?
6. L123. phytophthora? This is a generic name. Use capitalized and italicized spelling.
7. L121-132. This is not a GC-MS analysis. Please describe the experiment in methodology. What was the Control? Thiabendazole?
8. L135-140. this is not a description of mycelial morphology. The paragraph should be merged with the previous one.
9. L137. The authors did not perform a correlation analysis.
10. Fig.1. The title is not appropriate. The photograph shows linear growth of P. noxius on medium containing different concentrations of essential oil of Cunninghamia lanceolate var. konishii. Be sure to label the oil concentrations and control variants in the photograph. Was triflumizole used?
11 Fig. 2. The title is not appropriate. The photograph shows linear growth of P. noxius on medium containing different concentrations of cedrol. Be sure to label the concentrations and control variants in the photograph. Was triflumizole used?
12 Fig. 3. I do not see photographic documentation from the control variant.
13. L200. What does the value (n≧3) mean?
14. L223. Bacillus subtilis - put the Latin name in italics.
15. L281. Breda de Haan spores?
16. L365. It is not relevant information who isolated the fungus P. noxius. However, complete the description with important information such as the strain number, the collection in which it is stored and the origin of the strain including the place of isolation.
17. L368. Exudates? Please describe precisely what was tested in the experiment. What were the control variants? Describe precisely in how many replicates the study was conducted. What was measured? At what time was the research conducted? Under what conditions was the fungal culture conducted? I found in lines 27, 29 of the abstract and in the conclusion section of L512 that the reference compound for evaluating the antifungal properties of essential oil of Cunninghamia lanceolate var. konishii is triflumizole.
Author Response
Reviewer 1
Commend 1. The abstract should include brief information about the scope of the research work and the research methods used.
Response:
Thank you for your thoughtful suggestion. We have incorporated the sentences from L.23 to L.26 to courageously elaborate on the scope of this study.
Commend 2. Keywords: repeat the information contained in the title of the paper. This should be amended.
Response:
We value your suggestion, and have revised the keywords to include the following terms: Cunninghamia lanceolata, essential oil, brown root rot fungus, cedrol, antifungal activity, oxidative stress, and apoptosis. This adjustment aims to prevent redundancy with the title (L.42-43).
Commend 3. The title of section 2.1. reads like a summary. The title should be changed e.g. Antifungal activity of selected essential oils. The material and methods section should describe the experiment!
Response:
We appreciate your suggestion. In the revised version, we have adjusted the title to “Comparing the antifungal activity of Taiwan's five precious woods” (L. 91), as per your instructions.
Commend 4. L106. Again the title of section 2.2. is a conclusion?
Response:
In the revised version, we have modified the title to “Composition analysis of CLOL and evaluation of its potent antifungal activity compounds” (L. 155).
Commend 5. L116. GS-MS?
Response:
No, it is the “GC-MS”, we added the full name “gas chromatography-mass spectrum (GC-MS) in the revised version (L. 116 to 117).
Commend 6. L123. phytophthora? This is a generic name. Use capitalized and italicized spelling.
Response:
We apology our mistake it should be used the scientific name, it is P. noxius (L. 136).
Commend 7. L121-132. This is not a GC-MS analysis. Please describe the experiment in methodology. What was the Control? Thiabendazole?
Response:
Yes, we do describe the experiment in material and method section in our article, section 3.1. Preparation Essential Oil from Cunninghamia lanceolate var. konishii Wood and Its Composition Analysis.
Commend 8. L135-140. this is not a description of mycelial morphology. The paragraph should be merged with the previous one.
Response:
We have modified sentences to “with an increase in the dosage of COCL and cedrol, the growth of fungal colonies decreases. Ethanol, used as a solvent for dissolving cedrol, does not reduce the growth of fungal colonies compared to the control group; in fact, it may even promote strain growth. This indicates that ethanol does not affect hyphal growth, consistent with the antibacterial index results. To observe changes in hyphal morphology, this study employed scanning electron microscopy (SEM) at 2000x magnification for the essential oil and cedrol groups. As shown in Figure 3, with an increase in the dosage, there is a noticeable deformation of hyphae, and as the dosage increases, the deformation becomes more severe” (L.149 to 157). In the revised version.
Commend 9. L137. The authors did not perform a correlation analysis.
Response:
In this study, we used ethanol to dissolve cedar alcohol. Therefore, ethanol was added to the control group. In our observations, no damage to the hyphae was found due to the presence of ethanol.
Commend 10. Fig.1. The title is not appropriate. The photograph shows linear growth of P. noxius on medium containing different concentrations of essential oil of Cunninghamia lanceolate var. konishii. Be sure to label the oil concentrations and control variants in the photograph. Was triflumizole used?
Commentd 11. Fig. 2. The title is not appropriate. The photograph shows linear growth of P. noxius on medium containing different concentrations of cedrol. Be sure to label the concentrations and control variants in the photograph. Was triflumizole used?
Response:
We acknowledge and appreciate your suggestion. The figure legends have been revised to read as follows: “Inhibition effects of 10, 20, and 40 µg/mL essential oil of Cunninghamia lanceolata var. konishii against P. noxius,” and “Inhibition effects of 10, 20, and 40 µg/mL cedrol against P. noxius.” Additionally, we have included the labeling of treatments on each plate, including triflumizole, in the updated version. Thank you for bringing this to our attention.
Comment 12. Fig. 3. I do not see photographic documentation from the control variant
Response:
As pointed out by the reviewer, we did not present the appearance of the control group (untreated group) hyphae under 2000X SEM observation. This is because, at this magnification, the hyphae, due to their healthy growth (excessively dense), cannot be observed as individual and clear entities. However, the reviewer may refer to Figure 4, where healthy and intact hyphae (Blank) observed at 4000X are depicted.
Comment 13. L200. What does the value (n≧3) mean?
Response:
We sincerely apologize for the typo; it should be “n=3,” and we have corrected it in the revised version. Thank you for bringing it to our attention.
Comment 14. L223. Bacillus subtilis - put the Latin name in italics.
Response:
Yes, thank you very much for bringing it to our attention. We have corrected its format to italicize it.
Comment 15. L281. Breda de Haan spores?
Response:
It is the typo, the sentence is corrected to “Cell apoptosis can occur through intrinsic and extrinsic pathways. Cedrol stimulates a series of reactions in the cells of the P. noxius, leading to apoptosis” (L. 316-318).
Comment 16. L365. It is not relevant information who isolated the fungus P. noxius. However, complete the description with important information such as the strain number, the collection in which it is stored and the origin of the strain including the place of isolation.
Response:
Yes, we added the detail description for the brown-rot fungus used in this study. “The strain (Exfo00145) stored at the Plant Pathology and Microbiology Center, National Taiwan University Experimental Forest. Isolate originated from the Tropical Arboretum at the National Taiwan University Experimental Forest, Lutsao, Taiwan (L. 409-412). In the revised version.
Comment 17. L368. Exudates? Please describe precisely what was tested in the experiment. What were the control variants? Describe precisely in how many replicates the study was conducted. What was measured? At what time was the research conducted? Under what conditions was the fungal culture conducted? I found in lines 27, 29 of the abstract and in the conclusion section of L512 that the reference compound for evaluating the antifungal properties of essential oil of Cunninghamia lanceolate var. konishii is triflumizole.
Response:
Regarding our negligence, we sincerely apologize for not accurately describing the materials and methods. Each experiment was conducted in triplicate, and the results were averaged, with triflumizole used as the control group. We greatly appreciate the reminders from the revuewer, and we have already made the necessary corrections as described in the revised version.
“Antifungal assays were performed three times and the data were averaged. Different concentration of CLOL or cedrol were added to sterilized potato dextrose agar (PDA) to give 10, 20, and 40 µg/ml concentrations of essential oil and cedrol. Triflumizole (40 µg/ml) was used as the reference drug. The testing plates were incubated at 27 ± 2 °C. When the mycelium of fungi reached the edge of the control plate the antifungal index was calculated as follows: Antifungal index (%) = (1- Da/Db) × 100, where Da: diameter of growth zone in the experimental dish (cm), Db: diameter of growth zone in the control dish (cm)” (L. 414 to 425).

Reviewer 2 Report
Comments and Suggestions for Authors
The research described in the paper "Antifungal activity of cedrol from Cunninghamia lanceolate 2 var. konishii against Phellinus noxius and its mechanism" could be used also to study the antifungal activity of other compounds or extracts/essential oils against other unwanted and pathogenic fungi, both of agricultural and health interest. Although not all labs have the possibility of using electronic microscope, however many of the other methods described can be utilized in antifungal studies of conventional and natural substances.
Author Response
Reviewer 2
Commend: The research described in the paper "Antifungal activity of cedrol from Cunninghamia lanceolatevar. konishii against Phellinus noxius and its mechanism" could be used also to study the antifungal activity of other compounds or extracts/essential oils against other unwanted and pathogenic fungi, both of agricultural and health interest. Although not all labs have the possibility of using electronic microscope, however many of the other methods described can be utilized in antifungal studies of conventional and natural substances.
Response:
Thank you very much for the valuable feedback from the reviewer. In this study, various analytical techniques were employed, including observations using SEM, measurements of changes in mitochondrial membrane potential, analysis of oxidative stress, and the expression of apoptosis-related proteins. Through these methods, it was confirmed that cedrol can induce the cell apoptosis pathway leading to the death of the brown-root fungus. This result holds significant reference value in agriculture and can be widely applied and promoted

Round 2
Reviewer 1 Report
Comments and Suggestions for Authors
The manuscript has been revised according to the guidelines. I recommend that the paper be accepted in its current version.
I congratulate the authors on their interesting research.